# Genetically Encoded Fluorescent Indicators for Imaging Brain Chemistry

**DOI:** 10.3390/bios11040116

**Published:** 2021-04-11

**Authors:** Xiaoke Bi, Connor Beck, Yiyang Gong

**Affiliations:** Department of Biomedical Engineering, Duke University, Durham, NC 27708, USA; xiaoke.bi@duke.edu (X.B.); connor.beck@duke.edu (C.B.)

**Keywords:** genetically encoded fluorescent indicators, genetically encoded calcium indicators, genetically encoded neurotransmitter indicators, protein engineering, neural imaging

## Abstract

Genetically encoded fluorescent indicators, combined with optical imaging, enable the detection of physiologically or behaviorally relevant neural activity with high spatiotemporal resolution. Recent developments in protein engineering and screening strategies have improved the dynamic range, kinetics, and spectral properties of genetically encoded fluorescence indicators of brain chemistry. Such indicators have detected neurotransmitter and calcium dynamics with high signal-to-noise ratio at multiple temporal and spatial scales in vitro and in vivo. This review summarizes the current trends in these genetically encoded fluorescent indicators of neurotransmitters and calcium, focusing on their key metrics and in vivo applications.

## 1. Introduction

The mammalian brain supports sophisticated mental processes such as cognition and memory within complex circuits of interconnected neurons. While the canonical transmission of information starts with the initiation and propagation of voltage waveforms throughout a neuron’s surface, a substantial amount of neural communication between neurons relies on chemical transmission over a range of scales. Local membrane depolarization opens voltage-gated calcium channels, permitting an influx of calcium ions. The calcium ions then activate various vesicle transport proteins to induce neurotransmitter exocytosis. Neurotransmitters then reach their postsynaptic targets by traversing the extracellular space in multiple ways. Neurotransmitters crossing the synapse can target ligand-gated ion channels and cause an immediate conformational change, allowing fast and spatially confined synaptic transmission [1]. Neurotransmitters traveling via volumetric diffusion activate their downstream targets through a slow, long-range process [1]. Some neurotransmitters, such as acetylcholine, can activate multiple receptors and thus induce various spatial and temporal postsynaptic responses depending on the distribution of receptors in the target cells [1].

The interrogation of complex and diverse neural processes requires a detailed examination of how specific neuron types contribute to neural circuit functions. Such investigations have demanded tools that allow the noninvasive detection of neural activity with genetic specificity in vivo. This challenge is now partially met by recently engineered genetically encoded voltage indicators (GEVIs) and genetically encoded fluorescent indicators of brain chemistry. A recent review summarized the latest progress on the development of multiple categories of GEVIs in detail [2]. Here, we describe and assess different classes of genetically encoded fluorescent indicators of brain chemistry from two perspectives: (1) key metrics of their performance in vitro and (2) their capabilities to detect precise neural activities in vivo.

## 2. Advantages of Genetically Encoded Fluorescent Indicators

The combination of optical imaging and genetically encoded fluorescent indicators provides three distinct advantages in vivo: (1) neuron-type specificity, (2) cellular resolution, and (3) high temporal resolution. 

In conjunction with cell-type-specific promoters, targeting sequences, or animal driver lines expressing recombinase in specific cell types, genetically encoded indicators selectively express in targeted neural populations. First, lentivirus and adeno-associated viral (AAV) vectors are widely used as vehicles for gene delivery in neuroscience studies. Serotype and cell-type-specific promoters determine the transduction efficiency and targeting of transgene expression, respectively. Recent studies of viral vectors gave insight into serotype and promoter optimization, supporting the in vivo expression of genetically encoded indicators with high efficiency and neuron-type specificity [3,4]. Second, the expansion of animal driver lines expressing a recombinase, including zebrafish Tol2kit [5], GAL4 fly lines [6], Cre-lox [7,8], and FLP-FRT mice systems [9], have enabled more robust and sophisticated control of gene expression [10,11]. These advancements in viral vectors and transgenic systems have tailored the expression of genetically encoded fluorescent indicators and narrowly specified the detection of neural activity. Such genetic specificity is missing in traditional electrical and electrochemical methods that aggregate information from all neuron types. 

Genetically encoded fluorescent indicators report neural chemistry with high spatial resolution and reach in conjunction with optical imaging. Modern in vivo imaging methods, such as one-photon, multiphoton, and light-sheet microscopes, have enabled closed-loop neuroscience and the noninvasive recording of thousands of neurons at multiple spatial scales and depths [12,13,14]. Specifically, the combination of different types of optical imaging methods and genetically encoded fluorescent indicators revealed dynamic patterns of activity across broad regions or within small areas of interest in vivo [15]. Modern animal preparations can directly image the cortical surface (~1000 μm depth) [16,17] or relay light to and from deep brain regions (~3000 μm depth) within small animal models [18]. 

Finally, optical imaging has sufficient temporal resolution (~1800 Hz) to examine biochemical dynamics in neural circuits [14,19]. Although slower than electrical measurements, the millisecond kinetics of genetically encoded fluorescent indicators can still efficiently capture the comparably slower chemical transmission processes.

## 3. Development and Architectures of Genetically Encoded Fluorescent Indicators

Recent protein engineering methodologies, including rational design, directed evolution, and computational approaches, have greatly advanced the development of genetically encoded fluorescent indicators. First, early protein engineering relied more heavily on rational design, which grafts known mutations onto homologous sequences or generated novel mutations based on established knowledge of protein structure. For example, the incorporation of known mutations in the cpGFP component of GCaMP resulted in GCaMP1.6 [20], and the substitution of cpGFP with mRuby resulted in RCaMP [21]. However, such approaches require knowledge of protein structure and are low throughput. Second, the advancement in large-scale directed evolution, aided by medium- or high-throughput experimental pipelines, significantly accelerated indicator development. For example, site-directed random mutagenesis followed by cell screening based on medium-throughput imaging helped develop GRAB_Ach_ (GACh3.0) [22] shortly after the development of GACh2.0 [23]. Finally, large-scale directed evolution screens, using high-throughput experimental pipelines guided by machine learning, further accelerated directed evolution and explored the indicator fitness landscape. For example, a machine-learning-guided optimization of the binding pocket within an existing PBP-based indicator helped create the novel serotonin indicator iSeroSnFR. The advancement of machine-learning-guided protein engineering also accelerated the development of other genetically encoded tools for neuroscience such as channelrhodopsin [24]. These computational methods have the potential to accelerate all indicator development methodologies in the near future. When combined with computational methods that predict structure [25], computational methods that predict function could support either iterative development of existing sensor or de novo development of new sensor classes. 

Genetically encoded indicators typically consist of a sensing domain and optical reporters. These indicators fall into classes based on their sensing domains. This work reviews bacterial periplasmic-binding protein (PBP)-based indicators (Figure 1a,d), G-protein-coupled receptor (GPCR)-based indicators (Figure 1b,e), and calcium-binding protein-based indicators (Figure 1c,f). Both classes use two architectures to report the action of the sensing domain with a fluorescence change, either through a Förster resonance energy transfer (FRET) architecture or a circularly permuted fluorescent protein (cpFP) architecture. The FRET architecture flanks the sensing domain with a pair of complementary fluorescent proteins (FPs): one FP, the donor, has an emission spectrum that highly overlaps with the absorption spectrum of the other FP, the acceptor. A conformation change in the sensing domain induced by ligand binding displaces one FP from the other. At close proximity, the FRET acceptor will increase its fluorescence by absorbing a fraction of the energy that would otherwise be emitted as photons by the FRET donor. Because FRET indicators typically employ native versions of FPs, the indicators exhibit high brightness and photostability. 

On the other hand, single-wavelength indicators typically comprise a circularly permuted FP linked to the sensing domain (Figure 1d–f). Circular permutation of an FP fuses the FP’s original N- and C-termini and opens new termini close to the chromophore of the FP. Linking the new termini to the responsive portions of a neurotransmitter- or calcium-sensing domain couples the conformation of the sensing domain to the strength of interaction between the solvent and the chromophore, and subsequently to the indicator’s fluorescence intensity. Typically, the sensing domain in the binding state closes the new termini of the cpFP, protects the chromophore, and facilitates high fluorescence. Single-channel indicators typically support large relative fluorescence change at the expense of brightness in the resting state.

## 4. Genetically Encoded Neurotransmitter Indicators

### 4.1. PBP-Based Indicators

GltI (also known as ybeJ) is an *E. coli*-derived glutamate-binding PBP. Its function derives from its Venus Flytrap Domain (VFTD), which undergoes titular closing upon ligand binding. The first FRET architecture PBP indicators (FLIPE [26] and GluSnFRs [27]) coupled the GltI neurotransmitter-binding action to a cyan fluorescent protein (CFP) and a yellow fluorescent protein (YFP) (Figure 1a). Additional optimization of these indicators targeted two key position groups, both mutations in the GltI-sensing domain and truncations of the linkers between the GltI and the FPs expanded the glutamate dynamic range [28]. 

The first cpFP architecture of PBP indicators was iGluSnFR; the indicator inserted a circularly permuted enhanced green-fluorescent protein (cpEGFP) into a loop of the interdomain hinge region of GltI [29] (Figure 1d). Additional developments in this class diversified the palette and types of neurotransmitters detected. First, replacement of the cpEGFP with cpmApple created R-iGluSnFR1 [30], a red-fluorescent indicator with the potential to record glutamate activity with reduced autofluorescence and phototoxicity. Second, additional engineering of the PBP-sensing domain increased specificity for other neurotransmitters. Such evolution created iGABASnFR [31], iAChSnFR [32], and iSeroSnFR [33], respectively, sensing GABA, acetylcholine, and serotonin (Table 1). Much like the development of GluSnFR, the development of iGluSnFR first optimized key regions of the indicators, such as the sensing domain and linkers, with site-directed mutagenesis. Follow-up large-scale directed evolution screens, using high-throughput experimental pipelines guided by machine learning, broadly explored the protein landscape [30].

### 4.2. GPCR-Based Indicators

GPCRs are endogenous proteins in many species that bind neurotransmitters with high specificity and change their conformation after binding. Protein engineers have employed GPCRs as the sensing domain in multiple genetically encoded indicators to detect various neurotransmitters. The first GPCR/FRET-based norepinephrine indicator, α_2A_AR-cam, placed the donor of a CFP-YFP FRET pair at the third intracellular loop (ICL3) of the α_2A_-adrenergic receptor and the acceptor at the C-terminus of the receptor [34] (Figure 1b).

GPCR-based neurotransmitter indicators are also employed in the cpFP architecture (Figure 1e). A novel suite of six dLight1 variants, selected from the high-throughput screening of a random linker library, was the first demonstration of such allosteric indicators. These indicators reported dopamine dynamics over a broad range of physiologically relevant dopamine concentrations (4 nM–2.3 μM) and high dynamic range [35] (Table 1). A similar screening strategy that combined insertion site optimization, linker optimization, and affinity tuning generated a series of GRAB_DA_ indicators [37]. These indicators reported dopamine dynamics with similar kinetics and signal-to-noise ratio (SNR) as the dLight1 variants but improved on brightness and the response consistency over multiple cell types (Table 1). 

Additional efforts expanded the color palette of dopamine indicators and types of neurotransmitters detected by this indicator architecture. First, a combination of targeted point mutations and FP substitution diversified the spectra of dLight1 indicators by introducing yellow- and red-shifted dLight1 variants (YdLight1.1 and RdLight1, respectively) [36] (Table 1). These yellow/red indicators were spectrally separable from blue-light-activated indicators. A palette of indicators enabled the multichannel simultaneous imaging of calcium dynamics and synaptic dopamine release. Second, the high-throughput screening over cpFPs coupled to one of several GPCRs-sensing domains resulted in the development of GRAB_5-HT_, GRAB_NE_, and GRAB_Ach_. These indicators, respectively, detected serotonin, norepinephrine, and acetylcholine using the sensing domains from the serotonin 2C receptor, the α_2A_-adrenergic receptor, and the human muscarinic receptor 3, respectively [22,23,38,39] (Table 1).

## 5. Genetically Encoded Calcium Indicators

Calcium flux is a proxy for neural activity. GECIs report such dynamics and are one of the most mature approaches in neuroscience. The architecture of GECIs runs in parallel to the architecture of neurotransmitter indicators. GECIs typically consist of a calcium-binding domain, a binding domain target peptide, and a reporter element based either on a single FP or a FRET pair.

### 5.1. FRET-Based GECIs

The first widely used FRET-based GECIs were the Cameleon family, which inserted were calmodulin (CaM) and M13 (a calmodulin target peptide) in between either a BFP-GFP or a CFP-YFP FRET pair [40] (Figure 1c). Multiple cycles of engineering optimized the FPs and linkers within these indicators. The replacement of YFP with more modern Citrine and Venus FPs generated the yellow Cameleon series of indicators (YC2.x and YC3.x), which improved the brightness, pH stability, photostability, and Cl insensitivity [28,41,42,43,44,45,46,47] (Table 2). Additional optimization of the linkers either between CaM and the M13 peptide, or between the sensing components and FPs, further improved the sensitivity and calcium affinity of YC 2.6 and YC 3.6, resulting in the YC-Nano series of indicators [48,49] (Table 2). 

Modern engineering further expanded the spectral diversity of FRET-based calcium indicators. Green-red [50] and near-infrared [51] FRET-based Ca^2+^ indicators employed red-shifted FPs. Such indicators take advantage of decreased tissue scattering with a longer wavelength to enable high-resolution imaging at superior depths in scattering tissue. For example, one-photon light-sheet imaging of the near-infrared iGECI [51] can detect cellular and subcellular Ca^2+^ dynamics at 400 μm depth in acute brain slices.

Modern engineering also reduced interference between endogenous proteins and CaM-sensor components using two approaches. The first approach redesigned the binding interface between CaM and its target peptide to reduce the endogenous CaM binding of the sensor peptide. These redesigned Cameleons (D2cpv, D3cpv, and D4cpv) covered a 100-fold range in Ca^2+^ affinity and measured small changes in Ca^2+^ concentration [52] (Table 2). The second approach employed troponin C (TnC), a Ca^2+^-binding protein in cardiac and skeletal muscle that has less interference with the cellular regulatory protein network. TnC was the sensing domain in TN-L15 [53], TN-XL [54], TN-XXL [55], and Twitch [56]. The most recent of these indicators, Twitch, has Ca^2+^ affinity in the neurons’ physiological range and had a larger dynamic range than the dynamic range of YC 3.6 (Table 2).

**Table 2 biosensors-11-00116-t002:** Key metrics of selected genetically encoded calcium indicators.

Genetically Encoded Calcium Indicator	Ca^2+^-Binding Domain	Reporter Elements	Dynamic Range Δ*R*/*R*_0_ or Δ*F*/*F*_0_ (In Vitro Unless Otherwise Noted)	Affinity (*K*_d_) (In Vitro Unless Otherwise Noted)	Kinetics
FRET-based GECIs					
YC 2 [44]	CaM	ECFP/EYFP	1.8	100 nM	*τ*_d_ = 83 ms
YC 2.6 [45]	CaM	ECFP/cpVenus	6.6	95 nM	rise *T*_1/2_ = 185 ms|decay *T*_1/2_ = 2.31 s (in neuron)
YC 3.6 [45]	CaM	ECFP/cpVenus	5.6	250 nM	rise *T*_1/2_ = 214 ms|decay *T*_1/2_ = 0.4 s (in neuron)
YC-Nano15 [48]	CaM	ECFP/cpVenus	14.5	15.8 nM	rise *T*_1/2_ = 159 ms|decay *T*_1/2_ = 2.38 s (in neuron)
iGECI [51]	CaM	miRFP670/miRFP720	6	15 nM/890 nM	rise *T* = 0.70 s|decay *T* = 14 s
D3cpv [52]	CaM	ECFP/cpVenus	5.1	600 nM	rise *T*_1/2_ = 108 ms|decay *T*_1/2_ = 9.5s (in neuron)
D4cpv [52]	CaM	ECFP/cpVenus	3.8	60 µM	ND
TN-XXL [55]	TnC	ECFP/cpCitrine	3.3	800 nM	rise *T*_1/2_ = 80 ms|decay *T*_1/2_ = 1.6s (in neuron)
Twitch-2B [56]	TnC	mCerulean3/cpVenus	8	200 nM	decay *T*_1/2_ = 2.1 s (in neurons)
Single-fluorophore GECIs					
GCaMP 1 [57]	CaM	cpEGFP	4.5	240 nM	*τ*_d_ = 200 ms
GCaMP 1.6 [20]	CaM	cpEGFP	5	146 nM	*τ*_d_ = 260 ms
GCaMP 2 [58]	CaM	cpEGFP	5	840 nM	rise *T*_1/2_ = 95 ms|decay *T*_1/2_ = 480 ms (in brain slice)
GCaMP 3 [58]	CaM	cpEGFP	13.5	660 nM	rise *T*_1/2_ = 95 ms|decay *T*_1/2_ = 650 ms (in neuron)
GCaMP-HS [59]	CaM	Superfolder GFP	4 (in matured motor neurons)	102 nM	decay *T*_1/2_ = 0.92 s (in neuron)
Fast-GCaMP-EF20 [60]	CaM	cpEGFP	18	6.12 µM	decay *T*_1/2_ = 35 ms
GCaMP 5D [61]	CaM	cpEGFP	22	730 nM	*k*_on_ = 7.8 s^−1^ (measured at 670 nM)
GCaMP 5G [61]	CaM	cpEGFP	33	460 nM	*k*_on_ = 6.5 s^−1^ (measured at 670 nM)
GCaMP 6s [62]	CaM	cpEGFP	63	140 nM	*k*_on_ = 4.30 × 10^6^ M^−1^s^−1^|*k*_off_ = 0.69 s^−1^
GCaMP 6m [62]	CaM	cpEGFP	38	170 nM	*k*_off_ = 2.06 s^−1^
GCaMP 6f [62]	CaM	cpEGFP	52	380 nM	rise *T*_1/2_ = 74 ms|decay *T*_1/2_ = 400 ms (in neuron)|*k*_on_ = 9.44 × 10^6^ M^−1^s^−1^|*k*_off_ = 4.01 s^−1^
jGCaMP 7f [63]	CaM	cpEGFP	30.2	174 nM	*k*_on_ = 1.34 × 10^7^ M^−1^s^−1^|*k*_off_ = 5.86 s^−1^
jGCaMP 7s [63]	CaM	cpEGFP	40.4	68 nM	*k*_on_ = 2.15 × 10^7^ M^−1^s^−1^|*k*_off_ = 2.87 s^−1^ (fast) 0.27 s^−1^ (slow)
jGCaMP 7c [63]	CaM	cpEGFP	145.6	298 nM	*k*_on_ = 3.56 × 10^6^ M^−1^s^−1^|*k*_off_ = 2.79 s^−1^
jGCaMP 7b [63]	CaM	cpEGFP	22.1	82 nM	*k*_on_ = 1.6 × 10^7^ M^−1^s^−1^|*k*_off_ = 4.48 s^−1^
jGCaMP 8f [64]	CaM	cpEGFP	78.8	334 nM	*k*_off_ = 37.03 s^−1^
jGCaMP 8m [64]	CaM	cpEGFP	45.7	108 nM	*k*_off_ = 18.25 s^−1^
jGCaMP 8s [64]	CaM	cpEGFP	49.5	46 nM	*k*_off_ = 3.68 s^−1^
G-GECO 1.1 [65]	CaM	cpEGFP	26	482 nM	*k*_on_ = 8.17 × 10^15^ M^−n^s^−1^|*k*_off_ = 0.675 s^−1^ (n = 2.6)
G-GECO 1.2 [65]	CaM	cpEGFP	24	1.15 µM	*k*_on_ = 8.55 × 10^17^ M^−n^s^−1^|*k*_off_ = 0.7 s^−1^ (n = 3.0)
R-GECO 1 [65]	CaM	cpmApple	16	150 nM	*k*_on_ = 9.52 × 10^9^ M^−n^s^−1^|*k*_off_ = 0.752 s^−1^ (n = 1.6)
B-GECO 1 [65]	CaM	BFP	7	160 nM	*k*_on_ = 4.68 × 10^12^ M^−n^s^−1^|*k*_off_ = 0.490 s^−1^ (n = 2.0)
NIR-GECO [66]	CaM	mIFP	8	215 nM	rise *T* = 1.5 s|decay *T* = 4.0 s
XCaMP-G [67]	CaM	cpEGFP	80	200 nM	rise *T* = 80 ms|decay T = 190 ms
XCaMP-Y [67]	CaM	cpmVenus	115	81 nM	rise *T* = 65 ms|decay *T* = 210 ms
XCaMP-R [67]	CaM	cpmApple	20	97 nM	rise *T* = 30 ms|decay *T* = 200 ms

### 5.2. Single-Fluorophore GECIs

The first single-wavelength GECI to gain wide application was Camgaroo1 [68], closely followed by the development of Pericam [69], GCaMP1 [57], and Case [70]. These early indicators had Ca^2+^ affinity at the top end of the neurons’ physiological Ca^2+^ range. Iterative improvements have since focused on the GCaMP scaffold (Figure 1f), resulting in the large GCaMP family: GCaMP1.6 [20], GCaMP2 [58], GCaMP3 [71], GCaMP-HS [59], Fast-GCaMPs [60], GCaMP5 [61], the GCaMP6 series [62] (6s, 6m, and 6f), the jGCaMP7 series (7s, 7f, 7b, and 7c) [63], and the most recent jGCaMP8 series (8s, 8m, and 8f) [64]. Early engineering strategies improved brightness and sensitivity by optimizing the cpGFP, specifically by the incorporation of known (GCaMP1.6, GCaMP-HS) or novel (GCaMP2, 2.3) mutations in the cpEGFP component. Intermediate engineering strategies improved the dynamic range at the cost of Ca^2+^ affinity by incorporating novel mutations in the M13/cpEGFP (GCaMP2) and cpEGFP/CaM linkers (GCaMP5D, E, F). Later engineering strategies improved Ca^2+^ affinity, dynamic range, and kinetics by optimizing the Ca^2+^-sensing domain, which employed high-throughput screening and additional protein structure information. These mutations near the calcium- and peptide-binding interface of CaM iteratively increased the dynamic range (Table 2) and position the calcium affinity at the neuron’s physiological baseline (GCaMP3, GCaMP5G-K, GCaMP6, jGCaMP7). A second class of mutations also created a series of indicators with varying kinetics that sensed action potentials with a 30–240 ms duration (Fast-GCaMPs). Additional perturbations within the GCaMP6, jGCaMP7, or jGCaMP8 series maximized imaging fidelity in various in vivo imaging preparations that required high brightness, fast kinetics, high sensitivity to single-action potentials, or strong linearity.

More recent engineering strategies expanded the color palette of single-fluorophore GECIs, which improved spectral separability from green-fluorescent indicators and enabled dual-channel imaging. These strategies often started with FP substitution on GCaMP3’s scaffold, followed by additional random mutagenesis. First, the substitution of cpGFP with cpmApple and cpmRuby generated the red-shifted R-GECO and RCaMP series, respectively [21,65]. Second, more multicolor variants (G-GECO, R-GECO, B-GECO, and the recent NIR-GECO [65,66]; Table 2) in the GECO family were engineered using the high-throughput screening of a randomly generated GCaMP3 mutation library (Table 2). These GECO variants could simultaneously label multiple targets with different colors or image deep brain regions. However, these variants suffered from weak intensity and limited contrast for in vivo imaging. The development of the recent XCaMP series first replaced the M13 in GCaMP4.1 with CaMKKα (ckkap) and then incorporated novel mutations into both the linker sequences and the ckkap sequence (XCaMP-G) [67]. From there, the XCaMP series further expanded its color palette by mutagenizing the ckkap sequence in RCaMP2 (XCaMP-R) or the chromophore sequence (XCaMP-B and XCaMP-Y). This multicolor suite of GECIs optimized dynamic range and temporal fidelity (Table 2).

CaMPARI [72] and its successor, CaMPARI2 [73], formed a special class of single-fluorophore GECIs that employed circular permutations of photoconvertible FPs. This architecture produced different changes in fluorescent intensity between the unconverted state and the converted state. CaMPARI could mark active neurons that experienced simultaneously elevated calcium levels and blue-light excitation but over a brain region larger than the field of view of the microscope. These indicators have already helped highlight important, large populations of neurons in vivo [74,75].

### 5.3. Light-and-Calcium-Gated Transcription Factor System

GECIs discussed above directly observe Ca^2+^ concentration. The recently developed FLARE [76] employed similar designs as CaMPARI but drove gene expression in active cells in addition to labeling them. The FLARE system was a transcription factor tethered to the cell’s membrane by a CaM-binding peptide and a caged protease cleavage site. Blue-light excitation exposed the cleavage site, while high intracellular calcium recruited a CaM-bound protease to the cleavage site. The simultaneous gating of blue light and activity-induced calcium influx then released the transcription factor, initiating the expression of a desired transgene after the transcription factor migrated to the nucleus. In the future, this modular design could allow researchers to perform targeted genetic manipulations in active neurons beyond labeling. The direct manipulation of active neuron classes could support refined perturbation of neural circuits.

## 6. In Vivo Applications of Protein Indicators of Neurochemistry

The ever-improving functionality of genetically encoded indicators of brain chemistry has enabled in vivo examination of neuron-type-specific function. The cpFP architecture of both PBP- and GPCR-based indicators detects neural activity with high sensitivity and specificity. These indicators have already supported successful experiments in live animals. 

PBP-cpFP-based neurotransmitter indicators relay neural activity with large SNR in vivo. iGluSnFR, the oldest class of PBP-cpFP indicators, imaged the in vivo dynamics of glutamatergic transmission at cellular resolution in three model organisms [29]. Under one-photon illumination, it reported glutamate signals in the individual neurons of worms, which preceded and predicted postsynaptic calcium transients. Under two-photon illumination, it revealed the spatial organization of direction-selective synaptic activity in zebrafish optic tectum and tracked task-dependent single-spike activity in the mouse forelimb motor cortex. More recently, cpFP indicators of other neurotransmitters likewise have had a biological impact. iGABASnFR dissected the synchronization of volume GABA release from different populations of interneurons with interictal spikes and seizures in a mouse model of epilepsy [31]. iSeroSnFR connected bulk serotonin increases in the medial prefrontal cortex (mPFC) and basolateral amygdala (BLA) to cued fear conditioning. iAChSnFR detected the acetylcholine-acetylcholinesterase dynamics in both superficial cortical layers and deeper regions in response to isoflurane and ketamine/xylazine anesthesia.

GPCR-cpFP-based neurotransmitter indicators have matched the capabilities of PBP-based indicators in vivo. For example, under two-photon illumination, dLight1.1 and dLight1.2 indicators enabled the robust and chronic detection of relevant dopamine transients in multiple mouse behaviors [77]. In the mouse striatum, the dLight1 indicators tracked locomotion- and learning-induced changes in millisecond dopamine transients. In the mouse cortex, dLight1 indicators correlated dopamine transients to learning and motor control. Indicators from the GRAB family provided similarly new insights into in vivo neuromodulation in different scales of tissue. For example, GRAB_ACh_ (Ach3.0) enabled the visualization of compartment-specific acetylcholine signals in the Drosophila olfactory system, and bulk cholinergic dynamics during the sleep–wake cycle in mice.

Single-fluorophore GECIs have been prevalent in live animal experiments. The two-photon imaging of GCaMP1 visualized an odor-evoked activity map in the Drosophila brain [78]. The one-photon imaging of GCaMP2 visualized the glutamatergic transmission in the Drosophila larval neuromuscular junction (NMJ) [79]. Subsequent generations of GCaMP, starting with GCaMP3, further expanded the in vivo applications to the worm [80], zebrafish [81], and rodents [82]. These experiments chronically visualized ensemble-level activity with cellular resolution and tracked the recruitment of neurons into functional circuits during learning [83]. 

The cpFP class of GECIs also initiated investigations of information distribution, synaptic transmission, and plasticity within subcellular compartments. GCaMP6s expressed in the axonal boutons of neurons in the mouse primary visual cortex revealed a differentiated orientation and direction selectivity within projections targeting multiple higher-order visual areas [84]. GCaMP5 expressed in spines detected both single evoked or spontaneous synaptic vesicle fusion events at the Drosophila neuromuscular junction [85]. GCaMP6s expressed in dendrites helped detect branch-specific calcium spikes that supported the long-lasting potentiation of postsynaptic dendritic spines [86] and the formation of hippocampal representation of space [87]. The superior SNR of the recently developed jGCaMP7 series was distributed in various ways: it could concentrate on multiple small neuronal processes or capture up to thousands of neurons over millimeter fields-of-view and at higher speeds (up to ~160 Hz) [88,89,90].

The spectral diversity of single-fluorophore GECIs have allowed integrated optogenetics/imaging and multichannel imaging in vivo. For example, simultaneous neural activation via channelrhodopsin-2 (ChR2) and imaging via RCaMP reported Ca^2+^ transients of ChR2-evoked muscle contraction in *C. elegans* [21]. The promoter-driven, cell type-specific expression of XCaMP-R, XCaMP-Gf, and XCaMP-B supported the fiber photometry imaging of parvalbumin (PV)-positive neurons, somatostatin (SST)-positive neurons, and excitatory pyramidal neurons, respectively, in behaving mice performing an object investigation task [67]. 

The class of FRET-based indicators have found fewer applications in vivo due to their smaller dynamic range and lower SNR compared to peer cpFP-based indicators. The most successful live animal application for these indicators has been the imaging of action potential-induced calcium flux. YC2.6, YC3.6, D3cpv, and Twitch can all report action potentials in mouse cortical neurons in vivo [49,56,91]. However, they could not detect responses in smaller Drosophila neurons [92]. PBP-FRET and GPCR-FRET indicators have not fared as well in vivo. Although these indicators demonstrate sufficient affinity, their dynamic range is only one-tenth of the dynamic range of peer cpFP indicators. This low response often fails to rise above the extra noise induced by motion and background autofluorescence present in live animal preparations. To date, these indicators have all failed to break into the live animal imaging frontier.

## 7. Conclusions

In this review, we have presented a broad overview of the existing genetically encoded neurotransmitter indicators and GECIs. Compared with other methods for neural recording, genetically encoded indicators possess several advantages. Their genetic specificity, enhanced brightness, dynamic range, and SNR enable the large-scale recording of neural chemistry at multiple temporal and spatial scales in vitro and in vivo. Advances in protein engineering and high-throughput screening have accelerated the indicator optimization pipeline in the past two decades. These technical advances have engendered genetically encoded biochemical indicators that not only support existing dissections of neural circuitry underlying behavior, but also serve future explorations of brain chemistry. Based on current trends in neuroscience, wherein neural activation and in vivo imaging are often conducted simultaneously, one possible future research direction in genetically encoded indicator development is pushing the fluorescent indicators toward red or NIR wavelengths. This red-shift would provide various benefits over green or yellow indicators, including deeper imaging due to the red light’s decreased scattering, decreased phototoxicity, and increased compatibility with blue-light-activated optogenetic actuators. 

## Figures and Tables

**Figure 1 biosensors-11-00116-f001:**
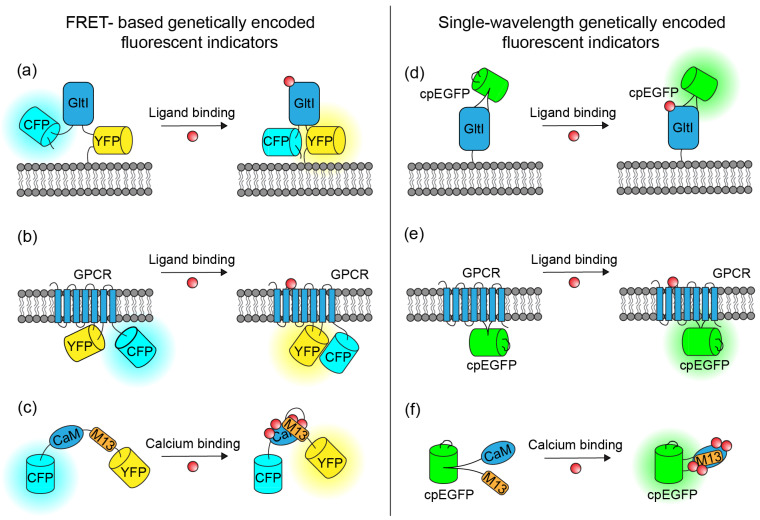
Multiple configurations of genetically encoded fluorescent indicators can image brain chemistry. (**a**) PBP/FRET-based genetically encoded neurotransmitter indicator (GluSnFRs). (**b**) GPCR/FRET-based genetically encoded neurotransmitter indicator (GPCR-cam). (**c**) FRET-based GECI (YC) (**d**) PBP/cpFP-based genetically encoded neurotransmitter indicator (iGluSnFRs). (**e**) GPCR/cpFP-based genetically encoded neurotransmitter indicator (dLight1). (**f**) cpFP-based GECI (GCaMP).

**Table 1 biosensors-11-00116-t001:** Key metrics of selected genetically encoded neurotransmitter indicators.

Genetically Encoded Neurotransmitter Indicator	Ligand	Reporter Element	Dynamic Range Δ*R*/*R*_0_ or Δ*F*/*F*_0_ (In Vitro Unless Otherwise Noted)	Affinity (*K_d_*) (In Vitro Unless Otherwise Noted)	On|Off Kinetics
PBP-based neurotransmitter indicators					
FLIPE [26]	Glutamate	CFP/Venus	ND	0.6 µM	*k*_on_ = 10.0 × 10^7^ M^−1^s^−1^|*k*_off =_ 60 s^−1^
GluSnFR [27]	Glutamate	CFP/YFP	0.07	150 nM	ND
SuperGluSnFR [28]	Glutamate	CFP/Citrine	0.44	2.5 µM	*k*_on_ = 3.0 × 10^7^ M^−1^s^−1^|*k*_off_ = 75 s^−1^
iGluSnFR [29]	Glutamate	cpEGFP	4.5	110 µM	*τ*_on_ = ~5 ms|*τ*_off_ = ~92 ms
R-iGluSnFR [30]	Glutamate	cpmApple	4.9	11 µM	ND
iGABASnFR [31]	GABA	cpSFGFP	2.5	9 µM	ND
iArchSnFR [32]	Acetylcholine	cpSFGFP	12	1.3 µM	*τ*_on_ = ~80 ms|*τ*_off_ = 1.9 s
iSeroSnFR [33]	Serotonin	cpSFGFP	8	310 µM	*τ*_on_ = ~0.5-10 ms (fast), ~5-18 s (slow)|*τ*_off_ = ~4 ms (fast), ~150 ms (slow)
GPCR-based neurotransmitter indicators					
α_2A_AR-cam [34]	Norepinephrine	CFP/YFP	−0.05	17 nM	*τ*_on_ = 40 ms
dLight 1.1 [35]	Dopamine	cpGFP	2.3	330 nM	*τ*_on_ = 10 ms|*τ*_off_ = 100 ms
dLight 1.2 [35]	Dopamine	cpGFP	3.4	770 nM	*τ*_on_ = 10 ms|*τ*_off_ = 100 ms
dLight 1.3a [35]	Dopamine	cpGFP	6.6	2.3 µM	*τ*_on_ = 10 ms|*τ*_off_ = 100 ms
dLight 1.3b [35]	Dopamine	cpGFP	9.3	1.7 µM	*τ*_on_ = 10 ms|*τ*_off_ = 100 ms
dLight 1.4 [35]	Dopamine	cpGFP	1.7	4 nM	*τ*_on_ = 10 ms|*τ*_off_ = 100 ms
YdLight 1.1 [36]	Dopamine	cpGFP V203Y/S72A	3.06	1.63 µM	ND/ND
RdLight 1 [36]	Dopamine	cpmApple	2.48	859 nM	*τ*_on_ = 14.1 ms|*τ*_off_ = 0.398 s
GRAB_DA1m_ [37]	Dopamine	cpEGFP	0.9	130 nM	*τ*_on_ = 60 ms|*τ*_off_ = 920 ms
GRAB_DA1h_ [37]	Dopamine	cpEGFP	0.9	10 nM	*τ*_on_ = 130 ms|*τ*_off_ = 1.9 s
GRAB_NE1m_ [38]	Norepinephrine	cpEGFP	2.3	930 nM	*τ*_on_ = 72 ms|*τ*_off_ = 680 ms
GRAB_NE1h_ [38]	Norepinephrine	cpEGFP	1.3	83 nM	*τ*_on_ = 36 ms|*τ*_off_ = 1890 ms
GACH 2.0 [23]	Acetylcholine	cpEGFP	0.76	2 µM	*τ*_on_ = 280 ms|*τ*_off_ = 762 ms
GACH 3.0 [22]	Acetylcholine	cpEGFP	2.8	2 µM	*τ*_on_ = 105 ms|*τ*_off_ = 3.7 s

## Data Availability

Data sharing not applicable.

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
