# Peer review of "Genetically Encoded Fluorescent Indicators for Imaging Brain Chemistry"

_biosensors, 2021, doi:10.3390/bios11040116_

Round 1
Reviewer 1 Report
The manuscript by X. Bi et al. reviewed the genetically encoded fluorescent sensors for brain imaging. The manuscript summarized the most widely used sensors based on their different architectures and briefly touched on their application in vivo. The manuscript will benefit from adding several aspects as detailed below:
- Recent literature reported 2 major GECIs that have gained tremendous interest, jGCaMP8 (doi:10.25378/janelia.13148243) and XCaMP (doi.org/10.1016/j.cell.2019.04.007). The authors should discuss these two types of sensors in this manuscript.
- The protein engineering rationale and methodology are largely omitted in the manuscript. The authors should discuss how modern technology such as directed evolution and computer-aided design has advanced sensor development. It would also be beneficial to discuss in the concluding remarks what are the advantage and disadvantages of protein engineering methods and how these methods impact trends for different sensor designs.
- Arguably voltage indicators may also belong to neurotransmitter sensors. They detect the voltage changes as a result of Na, K, H, Cl, etc. ion flow. The authors may want to briefly discuss these as well because these are excellent examples of FRET-based sensors that have been successfully adopted in vivo in multiple animal models.
- For concluding and future development sector, it is suggested to include special sensors that can record activity such as CAMPRI (doi: 10.1038/s41467-018-06935-2) and FLARE (doi: 10.1038/nbt.3909). This may be a new trend that will have a large impact on the field.
- Regarding in vivo application sector, the advancement of the optical instruments and virus technology should also be mentioned. Many of the sensors showed great performance largely due to special optical designs or certain serotypes of viruses.
- There seemed to be some typo in the second paragraph, such as page 2, line 51 citation, and line 53 ‘~3000 um’. If not a typo, the author should provide the exact literature where imaging of 3000 um has been achieved. The 3p imaging that the author cited only reached 1000 um imaging depth.
Author Response
Dear Editors and Reviewers,
We thank the editors and reviewers for the prompt, detailed, and insightful comments, criticisms, and suggestions on our manuscript. The reviewers have raised many excellent questions, and we have endeavored to address each issue in detail in this letter and in the revised manuscript. You will find the reviewers’ comments in italic blue font followed by our detailed responses in black font. Texts which have been newly added or altered in the revised manuscript are in red font. Accompanying this letter, we have also included the complete revised manuscript with the added and altered sections in matching red font, to facilitate your review of our revisions in context. We sincerely hope that you are satisfied with our responses. We thank the editors and reviewers again for your time and effort on reading and commenting on our manuscript.
Best Regards,
Xiaoke Bi, Connor Beck, Yiyang Gong

Reviewer 2 Report
In this manuscript, Bi and colleagues reviewed genetically encoded indicators based on the fluorescent proteins that are useful for the imaging of neural chemistry. They focused on the indicators based on the periplasmic-binding protein (PBP), G protein-coupled receptor (GPCR), and calcium-binding motif and explained their molecular design and properties. They covered a wide range of topics including both FRET and single fluorescent protein types of design and also imaging in vivo.
The manuscript is ready for publication after responding to the following revisions.
- Although the author is using the term “sensor”, this review contains the widely used acronym “GECI” includes the term “indicator”, a synonym of the “sensor”. I recommend unifying the term “sensor” to “indicator” through the manuscript.
- The unnecessary phrase “3/8/21 2:07:00 PM” exists at p.2 L51. That should be deleted.
- As the unabbreviated form of the FRET, the authors described “fluoresce resonance energy transfer” at p.2 L64. However, the FRET is defined as the acronym for Förster resonance energy transfer with IUPAC, thus that should be followed in this manuscript. Please see https://doi.org/10.1351/goldbook.FT07381 and https://doi.org/10.1351/goldbook.FT07385.
- Escherichia coli at p.2 L85 should be italic format.
- Although iSeroS-nFR is introduced at p.3 L100, it was not listed in Table1. That should be added to the table.
- IGECI at p.4 L157 must be iGECI.
- According to the nomenclature “cpGFP”, mApple and mRuby in the same sentence should be changed to “cpmApple” and “cpmRuby”, respectively, at p.4 L195. mApple in the item of R-iGluSnFR on Table 1 should be corrected to cpmApple.
- Reference 34 at p.11 L366 misses some information.
- There is a description of the FRET-based GECI, YC 3.0 in the p.6 L275. Probably the indicator YC 3.0 does not exist in the YC series. It had better be changed to YC or yellow Cameleon.
Author Response

(The authors gave the same response as above.)

Reviewer 3 Report
In this review, Bi et al. provide a comprehensive overview of the strategies that guide the construction of genetically encoded fluorescence sensors for neurotransmitters and calcium trafficking in the brain.
The work is well-structured, with a rational classification of the different mechanisms by which these sensors work and accurate examples for the various options. The bibliography is up-to-date and the writing clear. On the overall, the reader gets a fair description of the state of the art in the field.
On the overall, I feel the review is fit for publication.
A minor suggestion that I'd like to make is to add the reference numbering to the items in tables 1 and 2, so as to make such them easier to relate to the bibliography.
Finally, there is a typo in line 51.
Author Response

(The authors gave the same response as above.)

Reviewer 4 Report
The manuscript submitted to Biosensors entitled " Genetically encoded fluorescent sensors for imaging brain chemistry" by Xiaoke and co-workers presents a concise short review with an adequate number of references and information. The clear majority of relevant literature about this subject is referenced and properly discussed by the authors. This is a relevant topic with an increasing significance in recent years, carefully prepared, concise, and is a welcome addition to the literature. Therefore, this short revision has enough novelty/importance and should be considered for publication.
However, the conclusion would benefit if a personal point of view could be included. For instance, having in mind the most recent advancements what are the possible future research directions?
Thank you
Author Response

(The authors gave the same response as above.)

Round 2
Reviewer 1 Report
The authors have addressed all concerns raised by reviewers. The manuscript could be published as it is.